# Soluble PD-L1: From Immune Evasion to Cancer Therapy

**DOI:** 10.3390/life15040626

**Published:** 2025-04-08

**Authors:** Denisa Dragu, Laura Georgiana Necula, Coralia Bleotu, Carmen C. Diaconu, Mihaela Chivu-Economescu

**Affiliations:** 1Stefan S. Nicolau Institute of Virology, 030304 Bucharest, Romania; denisa.dragu@virology.ro (D.D.); coralia.bleotu@virology.ro (C.B.); carmen.diaconu@virology.ro (C.C.D.); mihaela.economescu@virology.ro (M.C.-E.); 2Faculty of Medicine, Titu Maiorescu University, 031593 Bucharest, Romania

**Keywords:** soluble, PD-L1, biomarker, immune evasion, cancer immunotherapy

## Abstract

Immunotherapy has emerged as a promising approach to cancer treatment, but only a small percentage of cancer patients benefit from it. To enhance therapeutic outcomes, it is essential to understand factors influencing immune response and tumor progression. Soluble PD-L1 (sPD-L1) has been identified as an essential element in immune regulation, with potential implications in cancer biology and treatment. This manuscript explores the sources and mechanisms of sPD-L1 production, its role in immune evasion and tumor progression, and its clinical significance. Elevated sPD-L1 levels have been linked to disease severity, survival, and treatment response in various malignancies, and as a consequence, strategies for combinatorial targeting of sPD-L1 with other immunotherapies are considered. Further studies are needed to understand sPD-L1 dynamics and to clarify the mechanisms of sPD-L1-mediated immunosuppression and its therapeutic implications.

## 1. Introduction

Immunotherapy is a relatively new treatment option primarily intended for patients with advanced, non-resectable, or metastatic cancer. It works by inhibiting immune checkpoints (ICKs) and activating the immune system to attack and destroy cancer cells. Blocking the PD-1/PD-L1 axis (programmed cell death proteins 1/programmed cell death proteins ligand 1) through immunotherapy, represented by the administration of ICK inhibitors (ICIs), has demonstrated great efficacy in the immunotherapeutic treatment of malignant melanoma, lung carcinoma, renal cancer, and gastric cancer, and remains the central way of targeting and activating the immune system.

Despite all of these advances, immunotherapy is not equally effective in treating all solid cancers. Some of these tumors, known as “immunologically cold tumors”, do not show an inflammatory infiltrate and have low levels of ICKs on their surface, which makes them resistant to immunotherapy. These tumors must be identified to be excluded from treatment with ICIs. In addition, immunotherapy comes with considerable costs. These costs include not only the drugs themselves, which can be extremely expensive, but also frequent doctor visits, lab tests, scans, and other procedures needed to monitor and manage treatment.

As a result, to support the rational and personalized use of these drugs, it is important to identify reliable biomarkers for highlighting and monitoring the effectiveness of immunotherapy.

Currently, combined positive score (CPS) determination by immunohistochemistry (IHC), microsatellite instability (MSI), and tumor mutational burden (TMB) are the only biomarkers approved by the Food and Drug Administration (FDA) used for selecting patients for ICI treatment [1]. Of these, CPS is the most used algorithm and is based on the determination of PD-L1 protein expression in tumor and immune cells using the immunohistochemistry (IHC) technique [2,3]. However, this method is time-consuming and may delay the start of therapy. Also, IHC analysis is often performed on archived samples or biopsies and may not reflect real-time PD-L1 levels within advanced tumors. Therefore, the discordance between the results of the tests and tumor heterogeneity affects the interpretation of PD-L1 analysis results. Thus, it is necessary to identify more reliable biomarkers to predict a patient’s response to immunotherapy treatment. Beside the cell membrane-bound PD-L1 (mPD-L1), several other extracellular forms have been identified, such as exosomal PD-L1 (exoPD-L1) and soluble circulating protein (sPD-L1). In the last five years, a soluble form of PD-L1 in the plasma of cancer patients has been reported as a potential predictive and prognostic biomarker in advanced cancer patients who received ICI treatment. The soluble ligand sPD-L1 circulating in the plasma can negatively regulate the function of effector T cells and can act as a decoy during treatment with anti-PD-L1 antibodies, reducing its effectiveness [4]. It can also serve as a prognostic or immune checkpoints predictive marker in various types of cancer. Therefore, elevated levels of sPD-L1 have been associated with reduced survival in esophageal, melanoma, lung, and gastric cancer [5,6,7], indicating its utility as a prognostic biomarker. Moreover, it can function as a marker for monitoring immunotherapy, the variation of its level in plasma indicating the adaptive response of tumor cells to the activation of T lymphocytes, and stratifying patients into responders or non-responders [8,9,10].

## 2. Sources of sPD-L1

The PD-L1 protein (also known as B7-H1) is codified by the *CD274* gene that is located on chromosome 9 and encodes a type I transmembrane glycoprotein that comprises an immunoglobulin Ig variable (V) domain, an Ig constant (C) extracellular domain, a transmembrane domain, and a cytoplasmic tail [11]. PD-L1 is constitutively expressed on both hematopoietic and non-hematopoietic cells, and cytokines such as interferon gamma (IFN-γ), tumor necrosis factor alpha (TNF-α), and interleukin-6 (IL-6) regulate PD-L1 expression. In addition to its localization at the level of the cell membrane, PD-L1 can be detected in the extracellular space, such as cell culture supernatant or human serum/plasma, and is known as sPD-L1 [12].

The sPD-L1 is produced and released primarily by tumor cells and mature dendritic cells (DCs), but not by immature DCs, macrophages, and monocytes. Myeloid cells and activated T lymphocytes show increased levels of mPD-L1, but sPD-L1 is generated particularly by myeloid cells, suggesting that distinct regulatory mechanisms might be involved in the production of the two forms of PD-L1 [13]. Moreover, high levels of sPD-L1 have been found in the supernatants of various PD-L1 positive cell lines, indicating that mPD-L1 on the cell surface is likely a source of sPD-L1. Notably, no correlation was found between tumor PD-L1 expression and sPD-L1 plasma levels in patients with diffuse large B-cell lymphomas, suggesting that the tumor microenvironment, including non-malignant cells, may also play a role in sPD-L1 production [14].

## 3. Mechanisms of sPD-L1 Production

Based on current data, various pathways of circulating PD-L1 production have been identified, including (1) spliced mRNA variants, (2) cleaved PD-L1 protein from cell surfaces, and (3) protein secreted by tumor cells through exosomes. Firstly, researchers have shown that sPD-L1 can originate from spliced variant mRNA lacking the transmembrane domain of PD-L1 [15]. Secondly, sPD-L1 may result from the matrix metalloproteinase cleavage of mPD-L1, potentially influencing immune responses [16]. Thirdly, cancer cells are known to secrete a form of PD-L1 via exosomes under the impact of immune cells and soluble molecular factors (e.g., IFN-γ expression) in various tumor tissues. Poggio et al. proposed that cancer cells secrete a significant amount of PD-L1 in exosomes, with only a fraction on the cell surface [17]. However, the soluble form of PD-L1, which is a free-floating protein, is distinct from the exosomal form exoPD-L1, a membrane-bound form associated with extracellular vesicles, and similar to the mPD-L1 form, which is also expressed on the surface of cells [12].

Current data support that proteolytic cleavage is the primary mechanism for generating soluble forms of PD-L1 [18]; however, a recent study suggested that the alternatively spliced CD274-L2A transcript forms as the predominant source of sPD-L1, at least in tumors and healthy samples [19]. These inconsistent results imply that the contribution of each mechanism may be tumor-specific, but additional studies are needed for a better understanding of these processes and their implications for therapy.

### 3.1. Generation of sPD-L1 by Alternative Splicing

Multiple PD-L1 spliced variants are produced by alternative splicing, the mechanism that processes pre-mRNA and generates various mature mRNAs, with different structures and functions (Figure 1).

In a study by Gong et al., two types of stably expressed sPD-L1 (PD-L1v242 and PD-L1v229-transmembrane domain–deficient) were identified in four patients experiencing NSCLC recurrence after anti-PD-L1 treatment. These variants act as negative regulators in anti-PD-L1 antibody treatment by competing with mPD-L1 [6]. In a similar study, four splice variants were identified in melanoma. Of these, three splice variants (PD-L1-1, PD-L1-9, and PD-L1-12) lacked the transmembrane domain and one variant loss of the intracellular domain, but all of them led to the secretion of sPD-L1, suggesting that the intracellular domain may stabilize PD-L1 on the cell surface. Moreover, the secretion level of sPD-L1 may rather reflect cytokine stimulation, as there can be discrepancies between the continuous expression of mPD-L1 and the subsequent increases in sPD-L1. The cytokines, such as IFNα, IFNγ, and TNFα, were shown to increase the splicing activities of PD-L1 leading to the secretion of sPD-L1 directly by tumor cells; thus, elevated levels of sPD-L1 were found to correlate with disease progression in a study on melanoma patients [20].

Another study identified a soluble form of PD-L1 that exhibits unique biochemical properties, having the ability to form a homodimeric structure. This secreted PD-L1 molecule has a distinctive 18-amino acid tail, which contains a cysteine residue that allows it to homodimerize, and in this form is more effective in vitro to suppress T cell proliferation and IFNγ production than monomeric sPD-L1 [15]. Interestingly, it was reported that the alternative splicing of the *CD274* gene, through the exaptation of an intronic LINE-2A retroelement, leads to the production of a truncated form of PD-L1. This isoform is produced in both healthy tissues and tumors, and, unlike the full-length PD-L1, sPD-L1 lacks inhibitory activity on T cells and acts as a receptor antagonist [19].

Various PD-L1 splice forms have been identified across multiple cancer types, contributing to immune modulation, yet their specific functions can vary significantly [21].

### 3.2. Generation of Soluble PD-L1 by Proteolysis

Another source of sPD-L1 is the proteolytic cleavage of mPD-L1 (Figure 2). Matrix metalloproteinases (MMPs) are among the proteases that contribute to the PD-L1 post-translational modifications. The MMP-7, MMP-9, MMP-10, and MMP-13 may cleave the mPD-L1 [16]. In a recent study it was shown that PD-L1 can be removed from the fibroblast surface by exogenous MMP-9 and MMP-13, and this process could limit their immunosuppressive capacity and lead to the exacerbation of inflammation in tissues [22]. Moreover, the cleavage of PD-L1 was evidenced in human oral squamous cell carcinoma cell lines OSC-19 and OSC-20. In this case, the MMP-7 and MMP-13 were able to degrade PD-L1, but the digested fragments of PD-L1 were not recognized by the anti-PD-L1 antibody [23]. Treatment with recombinant forms of MMP-7, MMP-9, or MMP-10 has been shown to enhance the production of sPD-L1, while the administration of recombinant MMP-9 or MMP-13 reduced the mPD-L1. Additionally, treatment with different MMP inhibitors against MMP-10 and MMP-13 led to an increase in the membrane-bound form of PD-L1 [22,23]. Proteolytic cleavage of the mPD-L1 can also be performed by the ADAMs (A disintegrin and metalloproteases), particularly by ADAM10 and ADAM17 who mediate the cleavage of mPD-L1 in malignant cell lines. The released sPD-L1 induces apoptosis in CD8 + T cells and reduces anti-tumor immunity [24]. Treatment with recombinant ADAM10 or ADAM17 resulted in a decrease in the membrane-bound form of PD-L1, and the inhibition of these proteases by GI254023X and TAPI-0 effectively prevented the formation of sPD-L1 [24,25,26].

## 4. Mechanisms Involved in sPD-L1-Mediated Immune Evasion and Tumor Progression

Even though the function of the full-length PD-L1, bound on the membrane of cells or exosomes, is well established, the biological activity of sPD-L1 is still poorly understood. It is well known that the PD-1/PD-L1 pathway is responsible for cancer immune escape and elevated PD-L1 expression correlates with disease stage and poor prognosis in many cancer types, such as melanoma, breast, ovarian, gastric, liver, kidney, and pancreatic [28].

Recently, numerous studies have focused on the identification and characterization of soluble forms of PD-L1 in vitro or/and in patients with various malignant diseases. In the majority of cases, sPD-L1 exhibits a similar mechanism of action as mPD-L1, by interaction with PD-1 receptors [29] leading to the inhibition of T lymphocyte-proliferation and down-regulation of the cell-mediated immunity. Also, it was shown that the exposure of CD4+ and CD8+ T cells to mature dendritic cell-derived sPD-L1 induced apoptosis in cells [13,30]. Moreover, a dimerized form of sPD-L1 was found to increase effectiveness in inhibiting T cell function compared to monomeric sPD-L1 [15].

However, it was shown that the immunosuppressive capacity of sPD-L1 is influenced by the binding affinity of the soluble isoforms to the PD-1 receptor. Recently, Liang et al. showed that the binding of high-affinity sPD-L1 variants, L3B3-hPD-L1 or L3C7-hPD-L1, to T cells PD-1 receptor suppressing T-cell activation attenuates the suppressive function of the T cells compared to low-affinity native PD-L1 [29]. In another study, research by Ng et al. revealed that the sPD-L1 isoform, produced through alternative splicing of the CD274 transcript known as CD274-L2A, lacks inherent T cell inhibitory activity. Instead, it functions as a PD-1 receptor antagonist, blocking the inhibitory effects of membrane-bound and exosomal PD-L1 [19]. Both studies reveal different splicing mechanisms that generate sPD-L1 variants that counteract immune suppression rather than promote it.

PD-L1 high levels can induce T cell inhibition by triggering multiple regulators of the cell cycle, including down-regulation of two important signaling pathways, PI3K-Akt-mTOR and RAS-MEK-ERK, involved in cell growth, differentiation, cell cycle progression, cell division, proliferation, and survival [31].

Since the up-regulation of PD-L1 in tumor cells seems to be a very important mechanism for immune evasion, this process can be activated by cancer cells using multiple signals such as elevated levels of transcriptional factors STAT3 and HIF-1α, oncogenic aberrant signaling pathways EGFR, MAPK, PI3k-AKT, several cytokines including IFN-γ, IFN-α, IL-2, IL-6, IL-10, IL-12, IL-15, IL-17, IL-25, IL-27, and even viral infections like EBV (Epstein–Barr virus) in the case of gastric and nasopharyngeal cancers [18].

Interestingly, in the case of cancer patients, both soluble molecules sPD-1 and sPD-L1 increase in plasma levels in order to regulate the effect of each other and to maintain the balance of the immune system. Under this circumstance, the overexpression of one will affect the function of the other. In other words, the overexpression of sPD-1 can prevent the T cell inhibition by blocking sPD-L1/mPD-L1 signaling, while the overexpression of sPD-L1 can inhibit T cell activation, supporting cancer immune evasion. There are several studies reporting an increase in sPD-1 levels after radiotherapy or even anti-PD-1 immunotherapy, suggesting that the main source of circulating sPD-1 is represented by T lymphocytes that are activated by these therapies. In advanced NSCLC (non-small cell lung cancer) patients treated with nivolumab (monoclonal IgG4 antibody that binds to PD-1 receptors), an increased or stable sPD-1 level was correlated with longer PFS (progression free survival) and OS (overall survival) while a high percent of NSCLC patients with increased levels of sPD-L1, treated with another anti-PD1, pembrolizumab combined with low doses of chemotherapy, exhibited progressive disease [32]. A different behavior was observed when the level of sPD-1 was analyzed in the pre-therapeutic plasma of different cancer patients. Higher levels of sPD-1 were found in HBV (hepatitis B virus)-related hepatocarcinoma (HCC) due to the activation of immune response against both HBV and tumor, and are associated with HCC patients’ OS [33], while in patients with pancreatic adenocarcinoma, increased levels of both sPD-1 and sPD-L1 were associated with worst survival [34].

These interactions can significantly impact the immune system, especially in relation to cancer, and lots of efforts have been made to understand the functional role of sPD-L1 in malignancy disease.

## 5. Clinical Importance of sPD-L1 in Cancer

### 5.1. sPD-L1 Impact on Cancer Progression and Prognostic Value

The levels of sPD-L1 in cancer patients have been found to correlate with disease severity, clinicopathological characteristics, survival rates, and treatment response. Thus, elevated levels of sPD-L1 consistently align with advanced disease stages, larger tumor sizes, and metastasis. Furthermore, higher sPD-L1 levels are associated with poorer outcomes and disease-free survival across various cancer types, like NSCLC, melanoma, esophageal, pancreatic, and gastric cancer [5,6,7,15,35,36,37,38] (Table 1). This correlation highlights the potential of sPD-L1 as a valuable prognostic indicator in oncology.

An increased expression of sPD-L1 was detected in the serum of patients with chronic hepatitis C (CHC), associated with its progression to HCC. The results of the in vitro studies suggest that the sPD-L1 elevated expression might be caused by the generation of PD-L1 in hepatocytes due to inflammatory cytokines, including IFN-γ [39]. Another study showed that in hepatitis B virus (HBV)-infected patients, the level of sPD-L1 was higher in the liver cirrhosis group, followed by the CHB and HCC groups, correlated with liver inflammation and liver enzymes [40]. Also, in HCC patients, high concentrations of sPD-L1 were associated with poor prognosis [41]. Recently it was shown that serum sPD-L1 levels are correlated with the PD-L1 expression in tumor cells [42], and the mechanism responsible for sPD-L1 expression seems to be related to the activity of the matrix metalloproteinase [29]. In this regard, inhibitors of MMP could also inhibit sPD-L1 generation [29]. In HCC, patients’ radiotherapy (RT) seems to increase sPD-L1 levels, resulting in a poorer short- and long-term efficacy, highlighting the possible therapeutic benefits of a combined strategy that includes RT and anti-PD-L1 immunotherapy [43].

In gastric cancer, we recently demonstrated that the concentration of sPD-L1 was significantly correlated with tissue PD-L1 protein, larger tumor size, advanced tumor stage, and lymph node metastasis. Additionally, high sPD-L1 levels (>103.5 ng/mL) were associated with significantly shorter OS than those with low levels (HR = 2.16, 95% CI 1.15–4.08, *p* = 0.017); however, multivariate analysis showed that sPD-L1 was not an independent prognostic factor for OS in this cohort [44]. Similar findings were reported by Kushlinskii N. E. et al. in a study of 101 patients, highlighting the potential role of sPD-L1 in gastric cancer prognosis, though it failed to demonstrate independent prognostic value in a multiparametric model [45]. However, a recent study by Shin et al. showed that a high pretreatment plasma sPD-L1 level was an independent prognostic factor in multivariate analysis for worse OS in advanced gastric cancer patients receiving first-line chemotherapy [46].

Although further studies are needed to clarify whether sPD-L1 can serve as an independent prognostic biomarker, its concentration may be a useful biomarker to assist with diagnosis, progression, and to evaluate the prognosis of cancer patients.

**Table 1 life-15-00626-t001:** The expression levels of sPD-L1 in various cancers and their relationship with disease prognosis and prediction of treatment efficacy.

Cancer Type	Findings/Prognosis	Patient Number	Reference
Esophageal squamous cell carcinoma	Soluble PD-L1 concentration is proportional to the expression of PD-L1 in tissue and is associated with a poor prognosis	73	[5]
Gastric cancer	Soluble PD-L1 concentration is proportional to the expression of PD-L1 in tissue; high sPD-L1 is correlated with tumor stage, tumor size, and is associated with a poor prognosis	85	[44]
Pancreatic adenocarcinoma	High sPD-L1 has poor prognostic significance	59 55	[34,47]
Hepatocellular carcinoma	sPD-L1 levels were positively correlated with PD-L1 expression in cancer cells; high sPD-L1 levels predicted poor outcomes	121	[42]
Clear cell renal cell carcinoma	Independent prognostic factor for survival	89	[48]
Urinary bladder cancer	Higher levels of sPD-L1 are associated with metastasis and poor prognosis	132	[49]
Soft tissue sarcomas	High sPD-L1 predicts metastasis and prognosis	135	[50]
Lung cancer	High sPD-L1 was significantly associated with worse OS (hazard ratio [HR] = 2.20; 95% CI: 1.59–3.05; *p* < 0.001) and PFS (HR = 2.42; 95% CI: 1.72–3.42; *p* < 0.001)	1188	[51]
Non-small cell lung cancer	sPD-L1 has been reported to correlate with advanced tumor stage, larger tumor size (>2.5 cm), lymph node metastasis, and distant metastasis	85	[52]
Glioma	Worse PFS and OS were observed in patients with higher baseline levels of sPD-L1 (*p* = 0.027 and 0.008, respectively)	60	[53]
Malignant melanoma	sPD-L1 level was associated with OS and PFS	30	[54]
Osteosarcoma	High sPD-L1 levels predict metastasis	70	[55]
Diffuse large B-cell lymphoma	sPD-L1 levels predict survival outcomes	164	[56]

OS—overall survival; PFS—progression-free survival.

### 5.2. sPD-L1 as a Guide for Therapy and Biomarker for Treatment Response

Currently, different immune cell populations and their activation states are being evaluated as potential biomarkers for predicting responses to checkpoint inhibitors. The proliferation of circulating PD-1+CD8+ T cells after anti-PD-1/PD-L1 therapies was observed in patients with solid tumors [57], while a higher number of peripheral CD14+CD16−HLA-DRhi monocytes before treatment was associated with a higher response rate in anti-PD-1/PD-L1 therapy [58]. Also, patients with a lower proportion of total Tregs within the CD4+ T cell population (CD4+CD25+CD127loFoxP3+) showed high toxicity after PD-1 inhibitor therapy [59]. Additionally, the neutrophil-lymphocyte ratio (NLR) was used to assess ICI treatment responses. A higher NLR has been associated with poorer survival outcomes in melanoma [60] and NSCLC patients [61]. However, these studies lack validation and additional research is necessary to establish their predictive value.

In this context, sPD-L1, a key protein regarded as a general marker of an inflammatory status [12], could become an indicator for guiding and monitoring cancer treatment, especially with ICIs involved. sPD-L1, released by cancer cells in truncated form, seems to be involved in immunosuppression but can also affect the efficacy of many treatment types, such as chemotherapy, radiotherapy, and immunotherapy, including PD-L1 blockade therapy [37].

The high expression of sPD-L1 in the bloodstream has been related to therapy results, since it can mirror the immunity status within the tumor microenvironment. For example, in melanoma, elevated sPD-L1 levels before treatment with ICIs were correlated with progressive disease [15,20]. More than that, a retrospective analysis of clinical data from 171 patients with advanced solid tumors who received nivolumab or pembrolizumab monotherapy revealed that patients with high sPD-L1 concentrations had a significantly poorer PFS and a tendency toward poorer OS compared with all other patients [62]. In patients with mesothelioma treated with the anti-PD-L1 durvalumab combined with the anti-cytotoxic T-lymphocyte antigen (CTLA)-4 tremelimumab, circulating sPD-L1 levels increased during therapy, supporting its role as a predictive biomarker of response to anti-PD-L1 therapy in cancer and the specific involvement of PD-L1 targeting in the release of its soluble form [36].

In this regard, a study reported an increase in sPD-L1 levels several weeks after ICI treatment, especially in the case of NSCLC or genitourinary cancer patients, and highlighted that the main source of sPD-L1 is the peripheral blood neutrophils. Another interesting observation is that the increase of sPD-L1 levels seems to be affected by the administration of tyrosine kinase inhibitors [63]. Hayashi and colleagues showed that increased sPD-L1 levels could reflect tumor PD-L1 expression and forecast how patients with NSCLC would respond to PD blockades [35]. Other studies reported that elevated pretreatment levels of sPD-L1 seem to predispose to cancer progression in malignant melanoma treated with checkpoint blockade, CTLA-4, or PD-1 inhibitors [20] while in advanced bladder cancer patients undergoing platinum and ICI therapy the pretreatment sPD-L1 levels are associated with a poor prognosis [64]. Likewise, Kim and team discovered that levels of sPD-L1 were predictors of how patients with extranodal NK/T cell lymphoma would respond to pembrolizumab treatment, showing how useful sPD-L1 is in treatment settings [65]. Furthermore, a systematic meta-analysis by Scirocchi F et al. that involved 1076 patients across various tumor types treated with immunotherapy indicated that elevated circulating sPD-L1 levels in cancer patients are associated with poorer survival, suggesting its potential as a prognostic biomarker for patient selection before immunotherapy [9].

Additionally, variations in sPD-L1 plasma levels could indicate the adaptive response of tumor cells to T lymphocyte activation and can be used in stratifying patients into responders or non-responders [10]. In a study on NSCLC patients treated with anti-PD-1 immunotherapy, dynamic changes were observed in sPD-L1 levels. Thus, after two cycles of therapy, sPD-L1 increased in 93% of the patients. Thereafter, sPD-L1 levels started to decline in responsive patients and continued to increase in non-responsive patients [8]. Similar findings were observed in colorectal cancer patients treated with regorafenib and the PD-1 inhibitor sintilimab. The progression of disease during combination immunotherapy correlated with a rise in sPD-L1 levels, while patients experiencing durable clinical benefit showed no significant change [10].

However, some solid tumors, such as prostate cancer, have been shown to be resistant to the treatment with anti-PD-L1 therapy [66]. The mechanism of therapeutic resistance among patients remains largely unknown. In view of the roles of sPD-L1 in tumor progression, we asked whether sPD-L1 could induce therapeutic resistance to anti-PD-L1 antibody treatment. In this respect, several studies have reported the primary cause of resistance to anti-PD-L1 therapy as being closely related to sPD-L1 splicing variants [6]. sPD-L1 seems to be involved in a resistance mechanism and can significantly influence the efficacy of antibodies used in therapy by acting as a decoy receptor for anti-PD-L1 antibodies. A recent study identified in two NSCLC patients a refractory to PD-L1 blockade therapy and two unique secreted PD-L1 splicing variants which lacked the transmembrane domain. These secreted PD-L1 variants worked as “decoys” of PD-L1 antibody in the HLA-matched coculture system of iPSC-derived CD8+ T cells and cancer cells. Moreover, by in vivo experiments it was demonstrated that only 1% of cells presenting the sPD-L1 variant can induce resistance to PD-L1 blockade [6,67]. Dimerized PD-L1-vInt4, another splicing variant of PD-L1, isolated from lung cancer patients can also function as a decoy for anti-PD-L1 antibodies and might affect the therapy results [68]. Nevertheless, future studies are needed to understand the behavior of soluble forms of PD-L1 in cancer biology and treatment settings.

### 5.3. Ongoing Challenges in Standardization of sPD-L1 Detection and Quantification

Although sPD-L1 is a promising biomarker, several technical challenges exist in its detection and quantification in plasma that is presently performed through immunoassays (ELISA, xMAP assay, etc.). These include variability in assay sensitivity, interference from similar molecules, and a lack of standardized protocols, which can lead to inconsistent results. Additionally, factors such as sample handling, storage conditions, and patient variability may influence sPD-L1 levels, underscoring the need for standardized methodologies to ensure reliable measurements.

Several studies have explored optimal sPD-L1 cutoff values as prognostic and predictive biomarkers. Our findings link sPD-L1 > 103.5 ng/mL to larger tumors, advanced stage, and shorter OS over three years [44]. Similarly, Ito et al. identified 50 pg/mL sPD-L1 as a risk factor for poor OS, correlating with older age and elevated CA19-9 and CRP but not tumor stage [69]. In esophageal squamous cell carcinoma, Shiraishi et al. reported higher sPD-L1 levels in stage IV (77.5 pg/mL) vs. stages I–III (44.8 pg/mL) [5]. Asanuma et al. found that sPD-L1 > 11.0 pg/μL in soft tissue sarcoma was linked to greater disease progression risk (41.8% vs. 20.7%, *p* = 0.013) [50]. Despite these efforts, no universal cutoff or standardized measurement protocol exists, as methods vary across studies (e.g., ELISA vs. IHC). Some studies suggest early increases in sPD-L1 during treatment may reflect immune activation and tumor destruction, while persistently high levels could indicate poor immunotherapy response.

Further validation is needed to establish clinically relevant thresholds, and we will expand on these aspects in our manuscript, addressing ongoing challenges in standardization.

## 6. Strategies to Block sPD-L1 Activity or Secretion

To counteract sPD-L1’s immunosuppressive effects, four primary approaches are under investigation. Soluble PD-1 (sPD-1) therapy involves administering recombinant sPD-1 to bind both membrane-bound and soluble PD-L1, preventing interactions with PD-1 on T cells and restoring anti-tumor immunity. Preclinical studies demonstrate that sPD-1 delivery via gene transfer or systemic administration disrupts PD-L1-mediated T-cell inhibition, including blocking interactions with B7-1 (CD80), which enhances anti-tumor immunity [18,70]. PD-L1 stability mechanisms are inhibited through targeting post-translational modifications such as glycosylation or palmitoylation. Enzymes like DHHC3/9 block PD-L1 palmitoylation, destabilizing it and lowering sPD-L1 levels [71]. Similarly, disrupting Sigma1 or FKBP51-mediated PD-L1 folding in the endoplasmic reticulum diminishes its secretion, enhancing T cell-mediated tumor clearance [72]. Small-molecule drugs like metformin (activating AMPK to degrade PD-L1) and curcumin (increasing ubiquitination) also destabilize PD-L1, reducing its soluble form [70]. Matrix metalloproteinase (MMP) inhibitors suppress sPD-L1 secretion by blocking proteolytic cleavage of membrane-bound PD-L1, as shown in experimental studies [16,73]. Targeting sPD-L1 isoforms focuses on splice variants like PD-L1–vInt4, which act as decoys; isoform-specific antibodies or RNA-based therapies could neutralize these variants [74]. Additional strategies include microRNA (miRNA) and small interfering RNA (siRNA) to suppress PD-L1 transcription/translation and nanoparticle-based delivery to remodel the tumor microenvironment (TME) and enhance therapeutic efficacy [75,76]. These approaches aim to disrupt sPD-L1’s role in immune evasion, either by neutralizing its activity, inhibiting its production, or bypassing its decoy effects.

## 7. Conclusions and Future Directions

sPD-L1 might be an important predictor of a cancer patient’s response to therapy by mirroring the level of PD-L1 within the tumor and the immune environment. It has emerged as a promising tool for early detection and treatment selection, providing real-time insights. It may serve as an effective tool for monitoring disease progression, assessing response to therapy, and detecting the potential development of treatment resistance, further supporting the personalized cancer management strategies.

This emphasizes the significance of comprehending its role in cancer treatment, particularly given the growing reliance on ICIs. Future prospective clinical trials should evaluate sPD-L1 utility as a real-time biomarker by regular assessment during the therapy, association with treatment responses, progression-free survival, and overall survival to identify subgroups who could benefit from specific therapies, particularly immunotherapies. Consequently, assessing pretreatment sPD-L1 levels may help predict the response to ICI therapy in patients with advanced solid tumors, potentially improving treatment efficacy and preventing unnecessary treatments. Advanced technologies such as liquid biopsies and high-throughput assays could sustain the real-time monitoring of sPD-L1 levels in combination with other biomarkers, shortening the time of the therapeutic decision-making and enhancing predictive accuracy for treatment outcomes. Moreover, combining therapies targeting checkpoint inhibitors and sPD-L1 could overcome decoy resistance mechanisms while the longitudinal monitoring of sPD-L1 will sustain the personalized treatment strategies in the oncology domain.

## Figures and Tables

**Figure 1 life-15-00626-f001:**
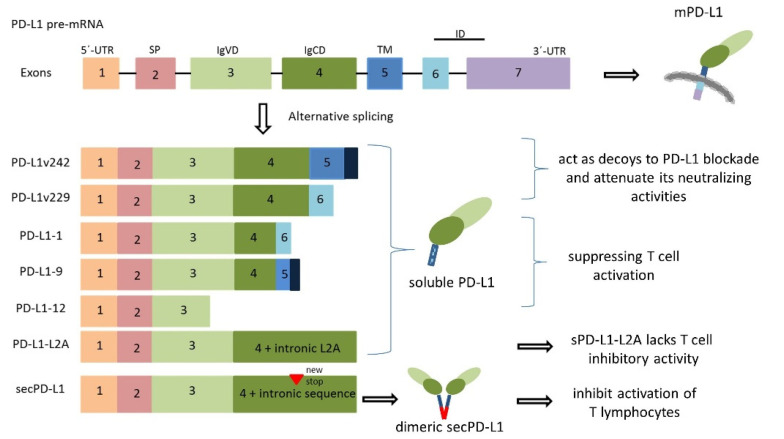
The PD-L1 splicing mechanism, soluble isoforms, and proteins functions. *PD-L1*/*CD274* full-length gene (**top**) consists of seven exons with exon 1 coding for the 5′ untranslated region (5′ UTR), exon 2—for the signaling peptide (SP), exon 3—for the IgV-like domain (IgVD), exon 4—for the IgC-like domain (IgCD), exon 5—for the transmembrane domain (TM), exon 6—for the intracellular domain (ID), and exon 7 encoding a part of the ID and a 3′ untranslated region (3′UTR). Soluble isoforms: PD-L1v242 and PD-L1v229—contain SP, IgVD, and IgCD domains: PD-L1v242 has a TM with deletion and extra amino acids at its C-terminal (dark blue) and PD-L1v229 also contains the ID [6]. The PD-L1-1 variant has a deletion in the ID. The splice occurs from the end of exon 4 to the middle of exon 6. PD-L1-9 has lost a region from exon 4 with a stop codon before the TM and an addition of two amino acids at the end (dark blue). The PD-L1-12 variant has a splice in the extracellular IgVD domain with a stop codon in exon 3. The resulting protein is truncated before the TM and terminates with a different amino acid [20]. The PD-L1-L2A variant contains the first four exons and an L2A LINE integration in intron 4, which acts as a terminal exon and polyadenylation signal [19]. secPD-L1 (**bottom**) contains the first 4 exons of PD-L1 and reads into the fourth intron which results in a new stop codon. The secPD-L1 splice variant has a unique cysteine-containing carboxyl-terminal domain which can dimerize [15]. The expressed proteins and their possible functions are also highlighted in the figure above.

**Figure 2 life-15-00626-f002:**
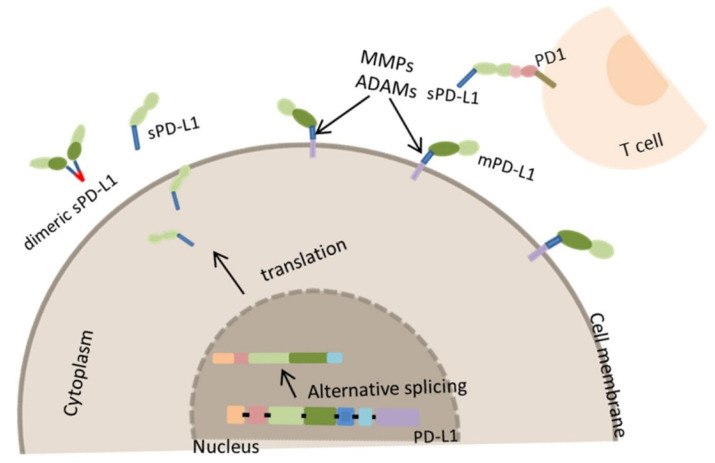
sPD-L1 protein cleavage from cell surfaces, along with its production via alternative mRNA splicing, contributes to the formation of sPD-L1 forms. The proteolytic cleavage of the mPD-L1 is accomplished by MMPs or by ADAMs. Proteolytic cleavage of the mPD-L1 can be performed by the matrix metalloproteinases such as MMP-7, MMP-9, MMP-10, and MMP-13 [16], and also by ADAM10 and ADAM17 [24]. These different forms of extracellular PD-L1 could be released into the cell culture supernatant or human blood. The PD-L1 immunoglobulin-like extracellular domain interacts with the immunoglobulin-like domain of the PD1 receptor and activates downstream signaling pathways, suppressing T cell activation [27].

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
