# Peer review of "Soluble PD-L1: From Immune Evasion to Cancer Therapy"

_life, 2025, doi:10.3390/life15040626_

Round 1

Reviewer 1 Report

Comments and Suggestions for Authors

Thanks for the opportunity to review this insightful and well-structured manuscript. Below are my detailed comments.

  • A more detailed discussion on the significance of sPD-L1 in hepatocellular carcinoma (HCC) and other liver-related malignancies would be valuable. You do mention HBV-related HCC, but it would help to elaborate on any unique mechanisms of sPD-L1 production in chronic liver diseases (e.g., cirrhosis, viral hepatitis) and how these might impact responses to immunotherapy. This would give the paper a broader clinical perspective for hepatologists.
  • The current manuscript covers multiple studies that used varying cutoffs and assays to quantify sPD-L1. It would be useful to include a short section discussing the ongoing challenges in standardizing sPD-L1 detection and quantification (ELISA vs. other assays). Noting whether any consensus protocols or reference standards exist—or are needed—would provide practical guidance for researchers and clinicians.
  • While you highlight the complexity of sPD-L1’s role in immune evasion and therapy response, the manuscript could benefit from a more robust “Future Directions” subsection. Suggest avenues such as:
    1. Prospective clinical trials evaluating sPD-L1 as a real-time biomarker.
    2. Combination approaches (e.g., targeting sPD-L1 alongside checkpoint inhibitors to overcome decoy resistance mechanisms).
    3. Longitudinal monitoring of sPD-L1 in various tumor types and correlation with clinical endpoints.
  • You nicely describe splice variants (e.g., PD-L1v242, PD-L1v229, PD-L1-9, etc.) and their roles as decoys. Some recent findings suggest that certain splice variants may lack inhibitory function (e.g., PD-L1-L2A) and act instead as receptor antagonists. Please consider emphasizing how these contrasting activities could be leveraged therapeutically—perhaps by designing therapies to block only the immune-inhibitory variants or by harnessing the non-inhibitory variants as natural antagonists.

Author Response

Thank you for your valuable comments. We have implemented the suggested corrections in the text, as outlined below. We hope these revisions have improved the manuscript's consistency and readability.

A more detailed discussion on the significance of sPD-L1 in hepatocellular carcinoma (HCC) and other liver-related malignancies would be valuable. You do mention HBV-related HCC, but it would help to elaborate on any unique mechanisms of sPD-L1 production in chronic liver diseases (e.g., cirrhosis, viral hepatitis) and how these might impact responses to immunotherapy. This would give the paper a broader clinical perspective for hepatologists.

R: Thank you for your feedback. We appreciate your suggestion and have incorporated a paragraph describing the mechanisms of sPD-L1 production and its impact on chronic liver diseases (pages 6-7, lines 251-264).

The current manuscript covers multiple studies that used varying cutoffs and assays to quantify sPD-L1. It would be useful to include a short section discussing the ongoing challenges in standardizing sPD-L1 detection and quantification (ELISA vs. other assays). Noting whether any consensus protocols or reference standards exist—or are needed—would provide practical guidance for researchers and clinicians.

R: Thank you for your suggestion! Indeed, standardizing sPD-L1 detection and quantification remains a major challenge due to the variability of methods used (ELISA, IHC, and other techniques). Currently, there are no consensus protocols or reference standards for sPD-L1 measurement, although multiple studies are underway to address this limitation. We included a new section addressing “Ongoing challenges in standardization of sPD-L1  detection and quantification” in the manuscript to provide clearer guidance for researchers and clinicians interested in using sPD-L1 as a biomarker (pages 9-10, lines 368-382).

While you highlight the complexity of sPD-L1’s role in immune evasion and therapy response, the manuscript could benefit from a more robust “Future Directions” subsection. Suggest avenues such as:

  • Prospective clinical trials evaluating.
  • Combination approaches (e.g., targeting sPD-L1 alongside checkpoint inhibitors to overcome decoy resistance mechanisms).
  • Longitudinal monitoring of sPD-L1 in various tumor types and correlation with clinical endpoints.

R: Thank you for your suggestion. We agree that a more detailed "Conclusions and Future Directions" section would enhance the manuscript. We have now expanded this section. We believe that these additions provide a clearer roadmap for the continued investigation of sPD-L1 and its implications (pages 10-11, lines 414-426).

You nicely describe splice variants (e.g., PD-L1v242, PD-L1v229, PD-L1-9, etc.) and their roles as decoys. Some recent findings suggest that certain splice variants may lack inhibitory function (e.g., PD-L1-L2A) and act instead as receptor antagonists. Please consider emphasizing how these contrasting activities could be leveraged therapeutically—perhaps by designing therapies to block only the immune-inhibitory variants or by harnessing the non-inhibitory variants as natural antagonists.

R: Thank you for your valuable feedback. We appreciate your suggestion and have now included a paragraph discussing several strategies to block sPD-L1 activity or secretion, including targeting sPD-L1. This addition enhances the discussion and provides a more comprehensive overview of potential therapeutic approaches (page 10, lines 384-405).

Reviewer 2 Report

Comments and Suggestions for Authors

This manuscript provides an overview of the role of soluble PD-L1 in immune evasion and its implications in cancer therapy. The discussion is well-structured and is supported by relevant literature. The insights into how soluble PD-L1 may act as both a prognostic biomarker and a potential therapeutic target add valuable depth to the field of cancer immunotherapy. I find the manuscript quite interesting and recommend this manuscript for publication after minor revisions.

Sections 1 and 2

  1. Since sPD-L1 levels are proposed as a biomarker, it would be helpful to understand potential measurement inconsistencies or confounding factors. It would be helpful for the readers if the authors briefly discussed any technical challenges or limitations in detecting and quantifying sPD-L1 in plasma.
  2. A short comparison with other checkpoint-related markers could give readers a broader perspective. For example, how does the role of sPD-L1 compare to other emerging immune biomarkers in predicting response to immunotherapy?
  3. The discussion on sPD-L1 as a biomarker is strong, but exploring whether blocking or modulating sPD-L1 could enhance immunotherapy outcomes would be interesting. But would it be possible to speculate on potential therapeutic strategies targeting sPD-L1 directly?

Sections 3 and 4

  1. The discussion of the different mechanisms of sPD-L1 production is well-structured, but the relative contribution of each pathway is unclear. Please provide a brief discussion of which of these mechanisms (alternative splicing, proteolysis, or exosomal secretion) plays a dominant role in specific cancer types.
  2. In the section where the interaction between sPD-L1 and sPD-1 in regulating immune responses is mentioned, could the authors discuss whether manipulating the balance between sPD-1 and sPD-L1 could serve as a potential strategy to enhance immune responses in certain cancers?
  3. The authors discuss sPD-L1’s immunosuppressive effects, but given the dual role of soluble immune checkpoint proteins, could there be any potential protective or anti-tumor functions of sPD-L1 in specific contexts?

Sections 5 and 6

  1. The discussion on sPD-L1 as a predictive biomarker for immunotherapy response is insightful. Would it be possible to expand on whether specific cutoff values for sPD-L1 levels could be used clinically to stratify patients into responders and non-responders?
  2. The paper highlights the dynamic changes in sPD-L1 levels during treatment, but it would be interesting to know whether fluctuations in sPD-L1 levels correlate with other immune biomarkers, such as cytokines or T-cell activation markers. Could the authors comment on whether there is any existing data on this?

Author Response

Thank you for your insightful comments. We have implemented the suggested corrections in the text, as shown below. We hope these revisions have enhanced the manuscript's coherence and readability.

Sections 1 and 2

Since sPD-L1 levels are proposed as a biomarker, it would be helpful to understand potential measurement inconsistencies or confounding factors. It would be helpful for the readers if the authors briefly discussed any technical challenges or limitations in detecting and quantifying sPD-L1 in plasma.

R: Thank you for your comment. We agree that discussing the potential measurement inconsistencies and confounding factors related to sPD-L1 as a biomarker is important. We included a new section addressing “Ongoing challenges in standardization of sPD-L1  detection and quantification” in the manuscript to provide clearer guidance for researchers and clinicians interested in using sPD-L1 as a biomarker (pages 9-10, lines 368-382). We believe this addition will provide a clearer understanding for the readers.

A short comparison with other checkpoint-related markers could give readers a broader perspective. For example, how does the role of sPD-L1 compare to other emerging immune biomarkers in predicting response to immunotherapy?

R:  Thank you for your suggestion. We included a paragraph discussing the role of sPD-L1 compared to other checkpoint-related markers (page 1, lines 41-44).

The discussion on sPD-L1 as a biomarker is strong, but exploring whether blocking or modulating sPD-L1 could enhance immunotherapy outcomes would be interesting. But would it be possible to speculate on potential therapeutic strategies targeting sPD-L1 directly?

R: Thank you for your feedback. We appreciate your suggestion and have incorporated a paragraph that discusses various strategies to inhibit sPD-L1 activity or secretion, including approaches targeting sPD-L1.

Sections 3 and 4

The discussion of the different mechanisms of sPD-L1 production is well-structured, but the relative contribution of each pathway is unclear. Please provide a brief discussion of which of these mechanisms (alternative splicing, proteolysis, or exosomal secretion) plays a dominant role in specific cancer types.

R: We appreciate your suggestion and have incorporated a paragraph that discusses these mechanisms (page 3, lines 99-104).

In the section where the interaction between sPD-L1 and sPD-1 in regulating immune responses is mentioned, could the authors discuss whether manipulating the balance between sPD-1 and sPD-L1 could serve as a potential strategy to enhance immune responses in certain cancers?

R: Thank you for your comment and suggestion. Indeed the balance between sPD-L1 and sPD-1 is a very important mechanism in regulating immune response. It is now emphasized in the paper (page 6, lines 212-291). Moreover, as you suggested we have included this approach in the new section entitled Strategies to Block sPD-L1 Activity or Secretion, page 10, lines 384-405).

The authors discuss sPD-L1’s immunosuppressive effects, but given the dual role of soluble immune checkpoint proteins, could there be any potential protective or anti-tumor functions of sPD-L1 in specific contexts?

R: We appreciate your suggestion and have incorporated a paragraph that discusses this subject (pages 5-6, lines 198-208).  

Sections 5 and 6

The discussion on sPD-L1 as a predictive biomarker for immunotherapy response is insightful. Would it be possible to expand on whether specific cutoff values for sPD-L1 levels could be used clinically to stratify patients into responders and non-responders?

R: Thank you for your comment! To date, there are no universally accepted cutoff values for sPD-L1 that allow for stratifying patients into responders and non-responders to immunotherapy. Available data suggest that high sPD-L1 levels may indicate a poor prognosis, and some studies have observed an early increase in circulating sPD-L1 during treatment, possibly associated with immune system activation and tumor destruction. However, persistently high sPD-L1 levels may signal a lack of response to immunotherapy. We will clarify these aspects in the relevant section of the manuscript (pages 9-10, lines 361-382).

The paper highlights the dynamic changes in sPD-L1 levels during treatment, but it would be interesting to know whether fluctuations in sPD-L1 levels correlate with other immune biomarkers, such as cytokines or T-cell activation markers. Could the authors comment on whether there is any existing data on this?

R: Thank you for your suggestion. We included a paragraph discussing the role of sPD-L1 compared to other checkpoint-related markers (page 8, lines 284-298).

Reviewer 3 Report

Comments and Suggestions for Authors

Please, see the attached document.

Author Response

Thank you for your insightful comments. We have implemented the suggested corrections in the text, as outlined below. We hope these revisions have improved the manuscript's consistency and readability.

  1.   What is the main research question or hypothesis guiding this review?  Kindly provide a precise definition of the primary research question.

R: Thank you for your comment. We have fully revised the abstract, focusing on highlighting the purpose of the manuscript and enhancing its clarity and structure (page 1).

  1.   Are there alternative interpretations regarding the function of sPD-L1 in immune evasion?

R: We appreciate your suggestion and have incorporated a paragraph that discusses this subject (pages 5-6, lines 198-208).

  1.   What is the comparative predictive and prognostic value of sPD-L1 relative to other immune checkpoint biomarkers?  Kindly provide additional comparative insights regarding sPD-L1 in relation to other immune biomarkers.

R: Thank you for your suggestion. We included a paragraph discussing the role of sPD-L1 compared to other immune biomarkers (page 8, lines: 284-298.

  1.     Can the paper address current clinical trials related to sPD-L1-targeted therapies?

R: Thank you for your suggestion. Although we could not identify any ongoing clinical trials specifically targeting sPD-L1, we have included a subsection that addresses this aspect, discussing relevant research and potential therapeutic strategies (page 10, lines 384-405).

  1.   What limitations in current research on sPD-L1 should be recognised?  What strategies could future research employ to overcome the limitations associated with current sPD-L1 measurements? Please include a discussion of potential experimental limitations in the current literature and enhance the conclusion to offer more robust insights for future research.

R: Thank you for your suggestion. We agree that a more comprehensive "Conclusions and Future Directions" section enhances the manuscript. We have expanded this section to offer a clearer perspective on the ongoing investigation of sPD-L1 and its implications (page 10-11, lines 414-426). 

Additionally, we have included a section “Ongoing challenges in standardization of sPD-L1  detection and quantification” addressing the limitations associated with current sPD-L1 measurements, where we discuss the lack of a universal cutoff or standardized measurement protocol, specifying that measurement methods vary across studies (e.g., ELISA vs. IHC), leading to inconsistencies in data interpretation (page 9-10, lines 361-382).

  1.     Some sources are outdated and lack strength; older references (prior to 2018) should be substituted with more recent studies to substantiate key claims.  Key claims lack appropriate citations.  It is essential to substantiate all significant claims with robust and current references.

R: We agree with your comment. However, we have included only the most necessary references regarding relevant studies before 2018. 

  1.     Enhance the logical coherence between sections, particularly in the transitions.

R: Thank you for your valuable suggestions. We have carefully reviewed the manuscript and made adjustments to improve its flow and readability.

Reviewer 4 Report

Comments and Suggestions for Authors

This review article covers biological effects of soluble PD-L1 (sPD-L1) in tumor pathogenesis, immune response, and immunotherapy and its role in cancer biology.

This specific strategy presented in this review is designed to describe other specific effects as potential prognostic and predictive biomarker for immunotherapy. It is evident that elevated levels of PD-L1 is related to disease severity, clinicopathological characteristics and recommendation of possible cancer therapy. The compiled data are supported with 2 figures and one table. The article concludes with 59 very recent literature references. This consolidated study constitutes important developments, which were never ever reported in such systematic and specific order and sequences.

The following suggested changes and recommendations should be introduced before the publication of the manuscript:

  • Page 2. Line 68. “The PD-L1 gene” It is biomarker (as stated in line 15) or gene? This must be clarified with supporting literature reference!
  • Page 2, Line 82. “Nonetheless, other possible sources cannot be excluded” Which are the other possible sources referred? Please, insert the literature reference.
  • Page 3, Line 119. Insert literature reference after “homodimeric structure”
  • Page 4, Figure 1 should be moved to line 104, (page 3) where is first mentioned.
  • Page 7. Table 1 legend, second column headline should be changed to “Finding/prognosis”
  • Page 9, Line 318. Conclusions. This section must be expanded. In its present format, the authors do not fully describe the desired/anticipated effects without citing the literature references. Authors should also include comparative data available in the literature              by inserting the references numbers specifically in the first paragraph in lines 319-324.

Additionally, the last sentence in line 329is not specific and authors should name the “future studies”, which are needed to understand the behavior …

The manuscript is of good quality, well-written, and meets the standard for articles published in Life. I recommend it for publication after the correction of these suggested changes.

Author Response

Thank you for your valuable comments. We have incorporated in the text the corrections that you suggested as presented below. We hope that now the manuscript has gained more consistency and readability.

  • Page 2. Line 68. “The PD-L1 gene” It is biomarker (as stated in line 15) or gene? This must be clarified with supporting literature reference!

R: Thank you for pointing this out. We clarified now this confusion (page 2, line 66).

  • Page 2, Line 82. “Nonetheless, other possible sources cannot be excluded” Which are the other possible sources referred? Please, insert the literature reference. 

R: That expression was a personal opinion, but we decided to remove it at the end (page 2, line 82)

  • Page 3, Line 119. Insert literature reference after “homodimeric structure”

R: The reference is number 15 and can be found at the end of the paragraph (page 4, line 146)

  • Page 4, Figure 1 should be moved to line 104, (page 3) where is first mentioned.

 R: Thank you for your suggestion. We moved the Figure where it was first mentioned (page 3, line 109).

  • Page 7. Table 1 legend, second column headline should be changed to “Finding/prognosis”

R: Thank you for your suggestion. We have modified it accordingly (page 7, line 282).

  • Page 9, Line 318. Conclusions. This section must be expanded. In its present format, the authors do not fully describe the desired/anticipated effects without citing the literature references. Authors should also include comparative data available in the literature              by inserting the references numbers specifically in the first paragraph in lines 319-324.

Additionally, the last sentence in line 329is not specific and authors should name the “future studies”, which are needed to understand the behavior …

The manuscript is of good quality, well-written, and meets the standard for articles published in Life. I recommend it for publication after the correction of these suggested changes.

R: Thank you for your comment. We have included references as suggested. We also agree that a more comprehensive "Conclusion" section enhances the manuscript. We therefore have expanded this section including “Future Directions” to offer a clearer perspective on the ongoing investigation of sPD-L1 and its implications (pages 10-11, line 414-426). 

Round 2

Reviewer 1 Report

Comments and Suggestions for Authors

Overall, these revisions further refine the manuscript. The organization is clearer, and the added sections address key gaps identified during the initial review. Thank you for your thorough and thoughtful improvements.

Reviewer 2 Report

Comments and Suggestions for Authors

Satisfied with the responses from the authors and the revisions. Accept the manuscript in its current form. 

Reviewer 3 Report

Comments and Suggestions for Authors

After significant revisions, I recomment for publishing.

Reviewer 4 Report

Comments and Suggestions for Authors

I read the response from authors and revised manuscript not once but twice and I am satisfied with all the important changes introduced into the manuscript. I strongly believe that in the current format the revised manuscript (and conclusions) is more clear and fully meet the all requirements for publication criteria as required by the journal.

As before, I am again enthusiastically recommend this important review article for publication as soon as possible.